# Biosensors Fabricated by Laser-Induced Metallization on DLP Composite Resin

Ran Zhang [1,2,†], Qinyi Wang [1,3,†], Ya Chen [1,4,†], Chen Jiao [1], Fuxi Liu [1,4], Junwei Xu [1], Qiuwei Zhang [1], Jiantao Zhao [1], Lida Shen [1,*] and Changjiang Wang [5]

1   Institute of Additive Manufacturing (3D Printing), Nanjing University of Aeronautics and Astronautics, Nanjing 210016, China
2   School of Advanced Technology, Xi'an Jiaotong-Liverpool University, Suzhou 215123, China
3   Nanjing Foreign Language School, Nanjing 210008, China
4   JITRI Institute of Precision Manufacturing, Nanjing 211805, China
5   Department of Engineering and Design, University of Sussex, Brighton BN1 9RH, UK
*   Correspondence: ldshen@nuaa.edu.cn
†   These authors contributed equally to this work.

**Abstract:** With the growing emphasis on medical testing, people are seeking more technologies to detect indexes of the human body quickly and at a low cost. The electrochemical biosensors became a research hotspot due to their excellent properties. In this study, dicopper hydroxide phosphate ($Cu_2(OH)PO_4$) was incorporated in resin, and the resin sheets were prepared by digital light processing (DLP). The copper base points were activated on the resin sheet surface by Nd: YAG laser and then covered by the electroless copper plating and the electroless silver plating. The laser could effectively activate copper base points on the resin surface. Furthermore, silver electrodes on the detection chips could distinguish glucose solutions of different concentrations well. Finally, a novel detection kit with a three-electrode chip was designed for rapid health testing at home or in medical institutions in the future.

**Keywords:** biosensor; DLP; laser activation; electroless plating; glucose testing

## 1. Introduction

With the development of technology and advancement in living conditions, people's awareness of kidney health is gradually increasing. As urine is the direct metabolite of the kidney [1,2], indicators from its tests can effectively reflect kidney health conditions. Traditional laboratory urine tests utilize automated urine analyzers [3,4], which have high equipment standards and require professionals to handle them. Novel biosensor technologies can efficiently analyze levels of metal ions [5], bacteria [6,7], glucose [8,9], and uric acid [10,11] to help patients form a quick judgment and are thus in high demand. Among them, electrochemical biosensors have the advantages of high accuracy, excellent selectivity, quick reaction, simple operation, and low cost. Some researches demonstrated the sensors exhibited better selectivity, sensitivity, and reproducibility [12–14]. They have been implemented in multiple fields, including disease monitoring [15,16], environmental surveillance [17,18], and food examination [19,20], with the most prominent applications in biology and medicine [21–23]. Electrochemical biosensors usually feature a double or triple electrode structure [24–27], using differences in electrochemical curves to characterize the content of substances in the sample [28–30], producing accurate results. Electrode materials are mainly platinum or gold for the working electrode and silver or silver chloride for the reference electrode [31–33]. In recent years, with the rapid development of biomonitoring and biosensing technology, electrochemical biosensors are evolving to become smaller-scale, multifunctional, digital, intelligent, and networked. Correspondingly, higher requirements for their manufacturing techniques have been put forward.

Traditional sensor structures (circuit substrates) often use subtractive manufacturing and produce circuits at the surface of the substrate by inkjet printing, optical printing, silkscreen printing, spraying, laser printing, and other methods [34–36]. However, such methods are time-consuming with long production periods, and are unsuitable for building customized structures [37,38]. Recently, additive manufacturing (AM) technology is being put into practice in automobiles, aeronautics and astronautics, printed electronics, and healthcare areas for its short production cycles and wide range of materials [39–42]. AM also offers item customization and high complexity. Moreover, compared to traditional manufacturing, AM can achieve functionalization through creating novel composite material and post-processing; examples include building a circuit at the surface of an AM prototype.

Further research has proved that circuits can be selectively built on the surface of AM prototypes [43–45]. For example, the selective in-situ growth method allows for a shorter process and the creation of particular patterns and has gained recognition among more researchers. Sun et al. [46] used an in situ growth method to deposit Ag particles directly onto the glucose oxidase (GOD) surface and used Nafion to fix them onto glassy carbon electrodes to produce Ag/GOD/Nafion composite glucose sensors that exhibited satisfactory performance. Our earlier research [47] proposes an original technique to metalize copper on ultraviolet photocuring resin selectively, but the susceptibility of copper to oxidization suggests that its stability does not satisfy the requirements for sensors. Therefore, we choose to use copper to induce silver growth in situ and manufacture electrochemical biosensors.

In this study, dicopper hydroxide phosphate ($Cu_2(OH)PO_4$) [48,49] was incorporated as an auxiliary material to ultraviolet (UV)-curing resin and molded the prototype by photocuring procedures. Then, the Nd: YAG laser was used to activate and reduce the copper ions to outline the pattern. Electroless copper plating and electroless silver plating were subsequently performed. A series of techniques were applied, including Scanning Electron Microscope (SEM), Energy Dispersing Spectroscopy (EDS), and X-ray Photoelectron Spectroscopy (XPS), to evaluate and analyze the surface element changes during the sensor manufacturing process. In addition, Cyclic Voltammetry (CV) was applied to characterize the electrochemical biosensors. Results demonstrated that the electrochemical biosensors had good sensing capabilities and could be used for urine testing. Finally, we presented an innovative and safe integrated testing device that can efficiently perform small-sample eco-friendly tests.

## 2. Materials and Methods

### 2.1. Preparation of Slurry and Digital Light Processing

In this study, the 10 wt.% dicopper hydroxide phosphate (Ningbo Yinzhou Puls Chemical Co., Ltd., Ningbo, China) was incorporated into the acrylic resin. All reagents were analytical grade. Then, the composite slurry was stirred for 30 min to homogeneously disperse $Cu_2(OH)PO_4$. The digital light processing (DLP) machine was applied to fabricate composite resin parts. Thus, the resin sheets ($30 \times 30 \times 2$ mm$^3$) were designed for experiments. The composite slurry was transferred to the forming cylinder and exposed layer by layer under UV light. The processing parameters are listed in Table 1. After the DLP processing, the resin sheets were further cured in a UV oven for 10 min, and then ultrasonically cleaned with ethanol and distilled water, respectively.

**Table 1.** The processing parameters of the composite resin sheets.

| DLP Processing Parameters | Contents |
|---|---|
| Cured layer thickness | 80 μm |
| Exposure time | 12 s |
| Light intensity | 10,000 mW/cm$^2$ |

### 2.2. Surface Modification

As shown in Figure 1, the as-prepared resin sheet was activated by the Nd: YAG laser, and the laser parameters were listed in Table 2. The patterned regions appeared on the resin surface after laser activation. Then, the copper and the silver were electroless plated on the activated surface. The chemical composition of the electroless plating solution was shown in Table 3. All reagents were analytical grade, and purchased from Shanghai Aladdin Biochemical Technology Co., Ltd., Shanghai, China. The electroless copper plating was applied to link the activated copper on the pattern because the copper on the laser-activated surface was discontinuous and non-conductive. The dense, homogeneous, and inert silver coating was formed by the electroless silver plating. Besides, the chemical reaction formulas for the electroless plating were as follows:

$$Cu^{2+} + 2H_2PO_2^- + 2OH^- \rightarrow Cu + 2H_2PO_3^- + H_2 \uparrow \tag{1}$$

$$2Ag^+ + C_4H_4O_6^{2-} + H_2O + 2NH_3 \rightarrow Ag_2O + (NH_4)_2C_4H_4O_6 \tag{2}$$

$$4Ag_2O + (NH_4)_2C_4H_4O_6 \rightarrow 8Ag + (NH_4)_2C_2O_4 + CO_2 + 2H_2O \tag{3}$$

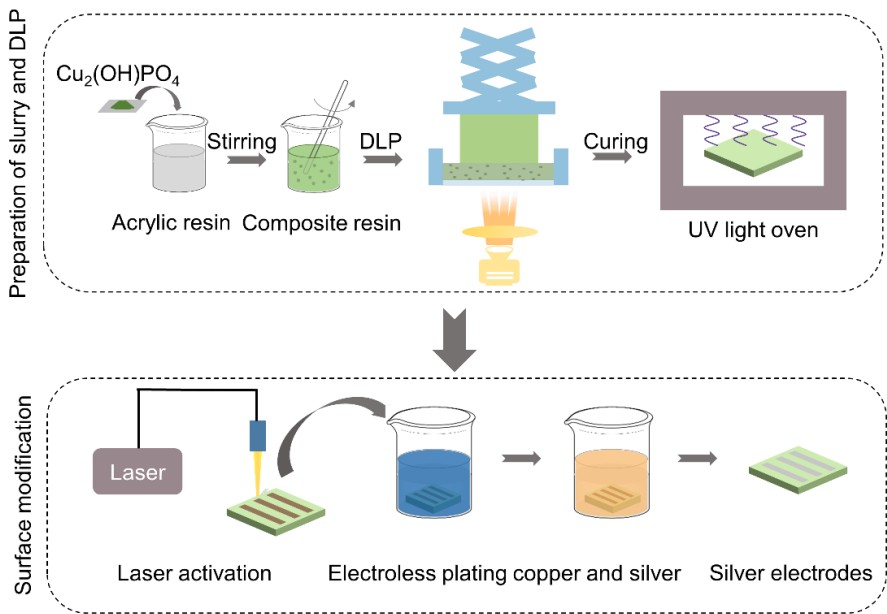

**Figure 1.** The schematic diagram of the preparation of silver electrodes.

**Table 2.** The laser parameters of the laser activation.

| Laser Parameters | Contents |
|---|---|
| Wavelength | 1064 nm |
| Pulse width | 100 ns |
| Spot diameter | 35 μm |
| Hatch distance | 50 μm |
| Scanning speed | 2500 mm |
| Pulse repetition frequency | 20 kHz |
| Laser fluence | 10.4 J/cm$^2$ |

**Table 3.** The composition of the electroless plating solution.

| Electroless copper plating | $CuSO_4 \cdot 5H_2O$ | HCHO | $HOCH_2COOH$ | $C_6H_{15}NO_3$ |
|---|---|---|---|---|
| | $C_6H_4SNCSH$ | | $C_{12}H_8N_2 \cdot H_2O$ | NaOH |
| | Time | 30 min | Temperature | 50 °C |
| Electroless silver plating | $AgNO_3$ | KCN | $C_4H_4Na_2O_6$ | $C_8H_4K_2O12Sb_2$ |
| | Time | 10 min | Temperature | 40 °C |

*2.3. Characterization*

The field emission scanning electron microscope (FE-SEM, Zeiss Ultra Plus, Oberkochen, Germany) was used to characterize the morphologies of the coating surfaces. The surface roughness was measured by the 3D surface profiler (Keyence VK-X1100, Osaka, Japan). Besides, the chemical composition and distribution of the coating surfaces were analyzed by the energy dispersive spectroscopy (EDS, Oxford Instruments X-MAX, Abingdon, UK), and X-ray photoelectron spectroscopy (XPS, Thermo Scientific ESCALAB 250Xi, Waltham, MA, USA). The cyclic voltammetry (CV) was conducted by an electrochemical workstation (Chenhua Instruments CHI660E, Shanghai, China). In addition, the silver electrodes were placed in a 10 M NaCl solution to determine the concentration of glucose at the scanning rate of 30 mV/s.

## 3. Results and Discussion

*3.1. Surface Morphology*

Figure 2 shows the surface morphology of composite resin, laser-activated surface, electroless copper plating, and electroless silver plating. In Figure 2a,e, the composite resin surface manufactured by DLP was relatively flat, and the $Cu_2(OH)PO_4$ particles were homogeneously distributed on the resin surface. The diameter of the particles was about 5 μm. In Figure 2b,f, laser-irradiated holes of approximately 50 μm were formed on the composite resin surface by the laser. In this study, the pulsed Nd: YAG laser was applied to activate the internal $Cu_2(OH)PO_4$ particles on the composite resin surface. In Figure 2c, the laser irradiation holes on the surface were filled with copper atoms formed by electroless copper plating on the laser-activated surface. However, the electroless copper coating cannot completely change the laser-activated resin surface, as shown in Figure 2g. In Figure 2e, the electroless copper plating was used to deposit copper on the laser-activated surface because the distribution of $Cu_2(OH)PO_4$ particles in the composite resin is discontinuous. Then, electroless copper plating formed a continuous electroless copper plating layer on the discontinuous activated copper layer surface. Finally, the dense silver coating was prepared on the electroless copper surface by electroless silver plating, as shown in Figure 2d,h. Since electroless copper plating was used as the pre-coating, the silver coating was evenly distributed on the surface, and the morphology of the laser-activated surface was significantly changed.

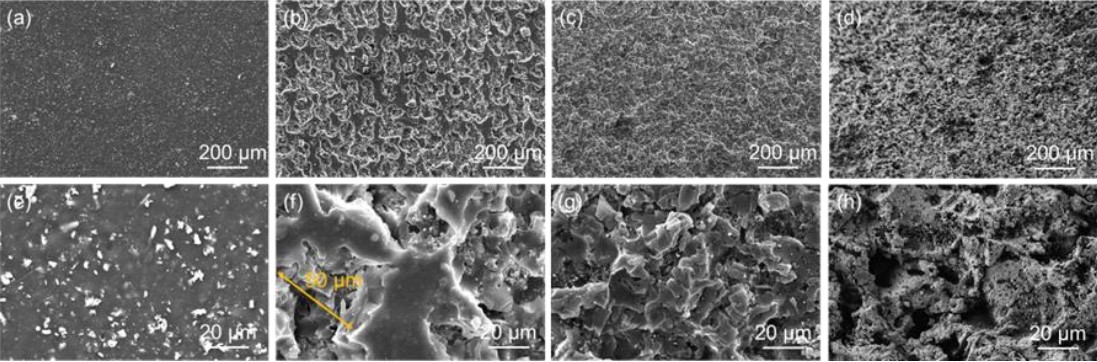

**Figure 2.** The SEM images of (**a**,**e**) composite resin; (**b**,**f**) laser-activated surface; (**c**,**g**) electroless copper plating; (**d**,**h**) electroless silver plating.

Figure 3 shows the cross-section of composite resin after electroless silver plating. In Figure 3a, the cross-section of the composite resin prepared by DLP shows a periodic cured layer with a thickness of 80 μm for each layer which is consistent with the cured layer thickness of DLP in Table 1. In Figure 3b at the 10 μm scale, the $Cu_2(OH)PO_4$ particles were observed to be homogeneously distributed in the composite resin and form a fragmented structure. It implied that $Cu_2(OH)PO_4$ could not be dissolved in photosensitive resin. Copper and silver prepared by electroless plating were attached to the surface of the composite resin and formed an 18 μm coating, as shown in Figure 3c. Different from the composite resin surface, the silver coating surface presented the cellular structure, which proved that the silver coating was well combined with the substrate and had no pores or cracks.

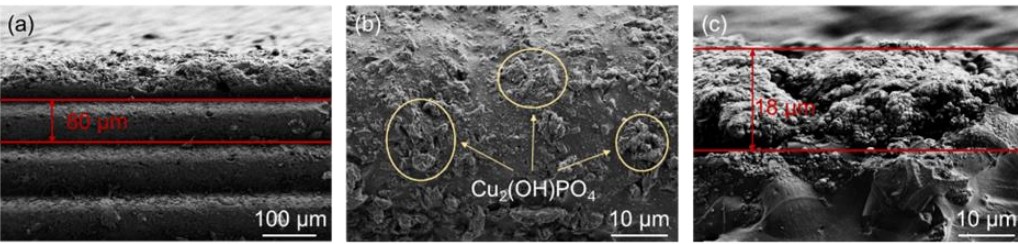

**Figure 3.** The SEM cross-sectional images of (**a**,**b**) composite resin; (**c**) silver coating.

Figure 4 shows the three-dimensional morphology of the surfaces. As shown in Figure 4a, the composite resin surface had small needle-like protrusions due to the accuracy error in DLP printing. Deeper pits appeared on the laser-activated surface in Figure 4b because the pulsed laser damaged the composite resin surface. In Figure 4c, the electroless copper plating process not only filled the pits but also further covered the protruding structures on the surface. In addition, Figure 4d shows the electroless silver plating area and the composite resin surface. The concave area was the edge of the laser-scanning pattern. It could be observed that the surface height above the concave area was lower than the surface height below the concave area because the upper area was the electroless plating area after laser activation, so it could be concluded that laser activation reduces the surface thickness. It was noted that the flatness of the upper area was better than that of the lower area. In Table 3, the surface roughness of the composite resin was 13.520 μm, which increased to 15.525 μm after laser activation. It was noted that the surface roughness of the electroless copper-plated surface and the electroless silver-plated surface decreased to 8.958 μm and 7.891 μm, respectively. It suggested that electroless copper plating and silver plating could improve the surface roughness by electroless deposition of metal. For electrochemical sensors, the electrical performance of the sensor is affected by the surface roughness of the substrate. Generally, the greater the surface roughness, the greater the resistance of the sensor. In addition, the surface uniformity of the sensor substrate can also improve the electrical performance of the sensor.

### 3.2. Chemical Composition

The chemical composition of the electrode surfaces was measured by EDS and XPS because it influenced the stability of the signal transmitted by the electrodes. Figure 5 shows the chemical composition distribution of laser-activated surfaces and electroless copper plating surfaces. Phosphorus was homogeneously distributed on the composite resin surface activated by the laser. However, the distribution of copper and oxygen was not entirely uniform. Copper was mainly distributed in the laser-activated region, and oxygen was mainly distributed near the region not activated by the laser. The results suggested that a small amount of copper was effectively activated on the surface of the composite resin through the laser activation process. However, the laser spot did not completely cover the surface, and the $Cu_2(OH)PO_4$ inside the composite resin was discontinuous. As a result, a continuous and conductive copper layer could not be obtained on the laser-activated

surface. Figure 5b$_1$,b$_2$ shows that the electroless copper plating covered uniformly the laser-activated surface. In addition, in Figure 5b$_3$,b$_4$, oxygen and phosphorus were also uniformly distributed on the surface, but the distribution density was lower than that of the laser-activated surface. It indicated that the electroless copper plating covered the activated surface evenly.

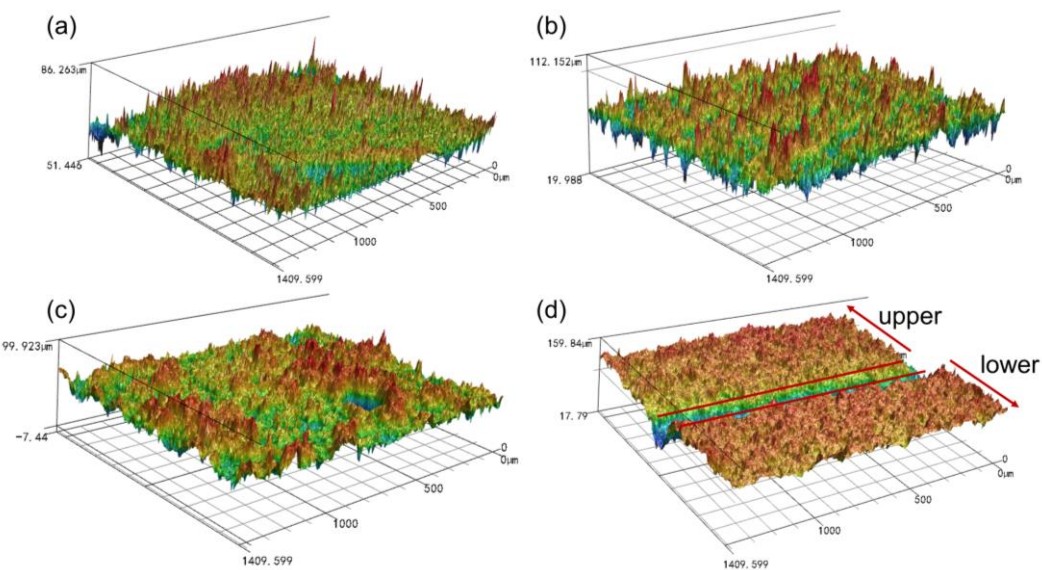

**Figure 4.** Three-dimensional morphology of (**a**) composite resin; (**b**) laser-activated surface; (**c**) electroless copper plating; (**d**) electroless silver plating.

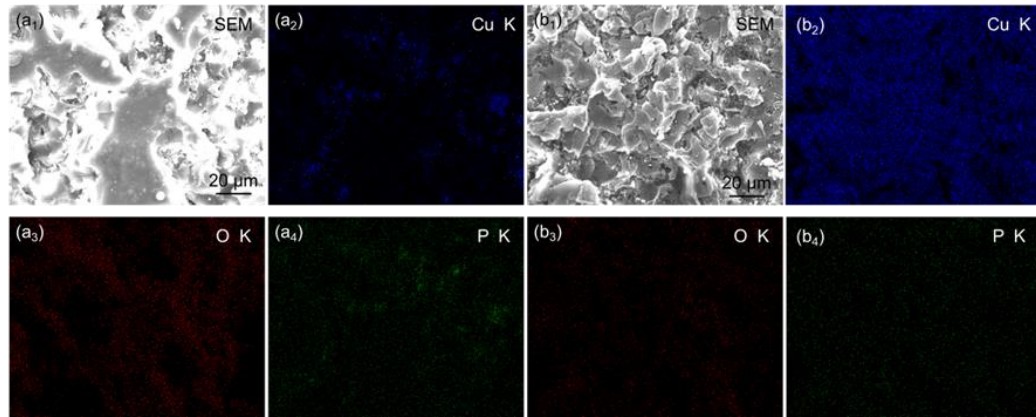

**Figure 5.** The EDS mapping of (**a$_1$**–**a$_4$**) laser-activated surface; (**b$_1$**–**b$_4$**) electroless copper plating.

Figure 6a$_1$–a$_5$ shows the chemical composition of the electrode surface after electroless silver plating. In Figure 6a$_2$–a$_5$, small amounts of oxygen and phosphorus were distributed on the surface of the silver plating, but large amounts of copper and silver were evenly distributed on the coating surface. It was noted that silver was mainly distributed on the coating surface, up to 58.87 wt.%, and copper reached 28.99 wt.%, as shown in Table 4. About 30 wt.% copper was considered in the copper layer that was not completely replaced during electroless silver plating. Figure 6b$_1$ shows the cross-section of the electrode after electroless silver plating. In Figure 6b$_2$, oxygen was mainly distributed in the composite resin because oxygen was one of the main chemical components of the resin. In Figure 6b$_3$,b$_4$, copper and silver were mainly distributed in the laser-activated surface area, but a small amount of copper was still distributed in the composite resin layer because copper existed as $Cu_2(OH)PO_4$ in the composite resin layer. As shown in Figure 6b$_5$, phosphorus was evenly distributed throughout the electrode cross-section, and $Cu_2(OH)PO_4$

was distributed not only in the composite resin layer, but also in a small amount in the metal electrode layer because the laser activation did not activate all the $Cu_2(OH)PO_4$.

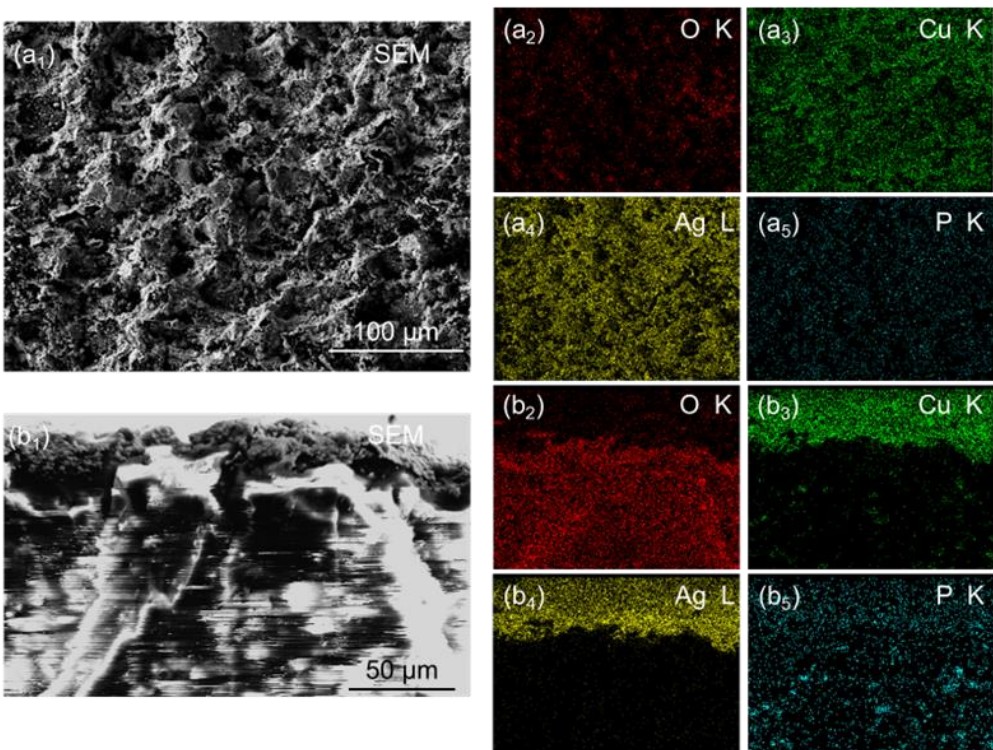

**Figure 6.** The EDS mapping of (**a₁–a₅**) silver coating; (**b₁–b₅**) cross-section of silver coating.

**Table 4.** The chemical proportion of the electroless silver coatings.

| Elements | Ag | Cu | O | P |
|---|---|---|---|---|
| (wt.%) | 58.87 | 28.99 | 11.99 | 0.14 |

In order to explore the activation effect of laser on the $Cu_2(OH)PO_4$ in the composite resin, XPS was carried out to detect the chemical composition of the composite resin surface and the laser-activated surface. Figure 7 shows the chemical composition of the composite resin surface and the laser-activated surface. In Figure 7a, phosphorus, carbon, and oxygen can be detected on the surfaces, but copper was only detected on laser-activated surfaces. As shown in Figure 7b,c,e, the carbon, oxygen, and phosphorus peaks of the two surfaces were similar. According to Figure 7b,c, the carbon on the surface mainly consisted of three bonds: C-C/C-H, C-O-C, and O-C=O, which were 284.8 eV, 286.3 eV, and 288.82 eV, respectively. These bonds mainly came from acrylic acid used to prepare composite resin. In addition, the oxygen peak was mainly composed of C-O and $Cu_2(OH)PO_4$ at 531.68 eV and 532.93 eV, respectively. In Figure 7e, the phosphate peaks of both surfaces were approximately at 133 eV, so phosphorus mainly came from metal phosphate ($Cu_2(OH)PO_4$). As shown in Figure 7d, the peak of copper was not detected in the composite resin because only copper metal or copper oxide could be detected near the copper peak by XPS. Copper in the composite resin mainly existed in the metal phosphate, so the $Cu_2(OH)PO_4$ could be found in the composite resin near the phosphorus peak. In addition, the copper peaks on the laser-activated surface were distributed at 932.18 eV, 934.81 eV, and 952.29 eV, respectively. In addition to two different hybrid copper elements (Cu $2p_{3/2}$ and Cu $2p_{1/2}$), copper also existed in the form of copper hydroxide, mainly from the by-product of laser activation.

Therefore, according to the XPS test results, it could be inferred that the chemical reaction under laser activation is:

$$Cu_2(OH)PO_4 + 2H_2O \rightarrow Cu + Cu(OH)_2 + H_3PO_3 + O_2 \uparrow \tag{4}$$

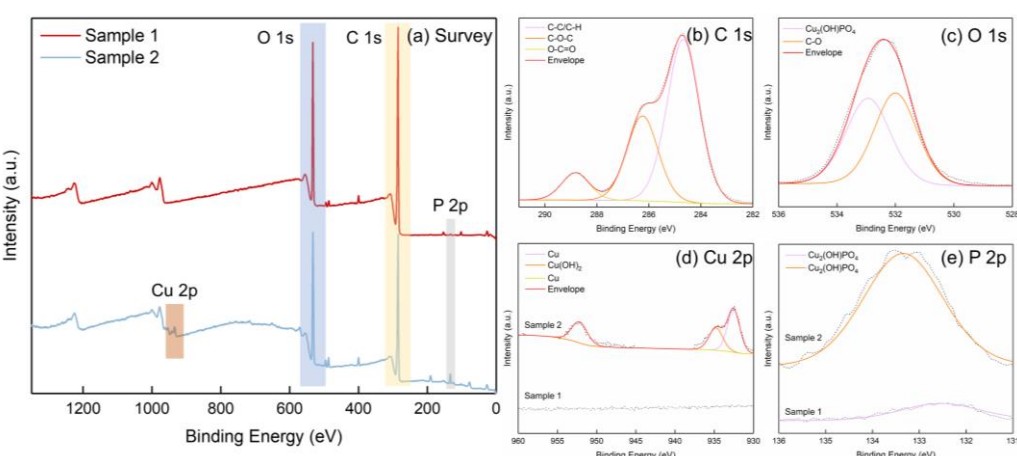

**Figure 7.** XPS spectra of composite resin and laser-activated surface: (**a**) XPS survey spectrum; (**b**) high-resolution C 1s spectra; (**c**) high-resolution O 1s spectra; (**d**) high-resolution Cu 2p spectra; (**e**) high-resolution P 2p spectra.

### 3.3. Cyclic Voltammetry and Applications

We aimed to fabricate an electrochemical biosensor that can be used for urine detection. We designed and manufactured the working electrode (WE), counter electrode (CE), and reference electrode (RE) made of silver on a prototype based on the previous manufacturing method to form a three-electrode electrochemical biosensor for urine detection. High urinary glucose levels may indicate the presence of diabetes mellitus. We have experimentally characterized the electrochemical biosensor using cyclic voltammetry (CV). For the assay, three electrodes were connected separately to an electrochemical workstation to perform the electrochemical measurements.

As a proof-of-concept demonstration, the different concentrations of glucose solution were used instead of urine. The different amounts of anhydrous glucose powder were added to 1 mol/L NaCl solution to create a concentration gradient, and 1 mol/L NaCl solution without glucose was used as a blank control. The CV was recorded and calculated as the normalized peak-to-peak current change (NPPCC), which was defined by Equation (5):

$$NPPPCC = \frac{I_b - I_a}{I_b} \tag{5}$$

where $I_a$ and $I_b$ are the inter-peak currents in the presence and absence of glucose respectively.

The CV curves for the real-time cyclic voltammetry assay with different concentrations of glucose were shown in Figure 8. Firstly, CV curves can be obtained at negative potential values and positive potential values because the reaction on the reversible electrodes is reversible. Then, CV is a vital method to study the electrode reversibility, and single metal electrodes are usually reversible electrodes. Thus, the silver electrode is a reversible electrode. All peak current values were normalized to the peak current of the NaCl solution without glucose. After the blank assay, the electrodes were placed sequentially into different concentrations of glucose solutions. The experimental results illustrated that an increase in glucose concentration led to a decrease in the CV peak current. So that we can characterize the concentration of glucose by the magnitude of the peak current measured.

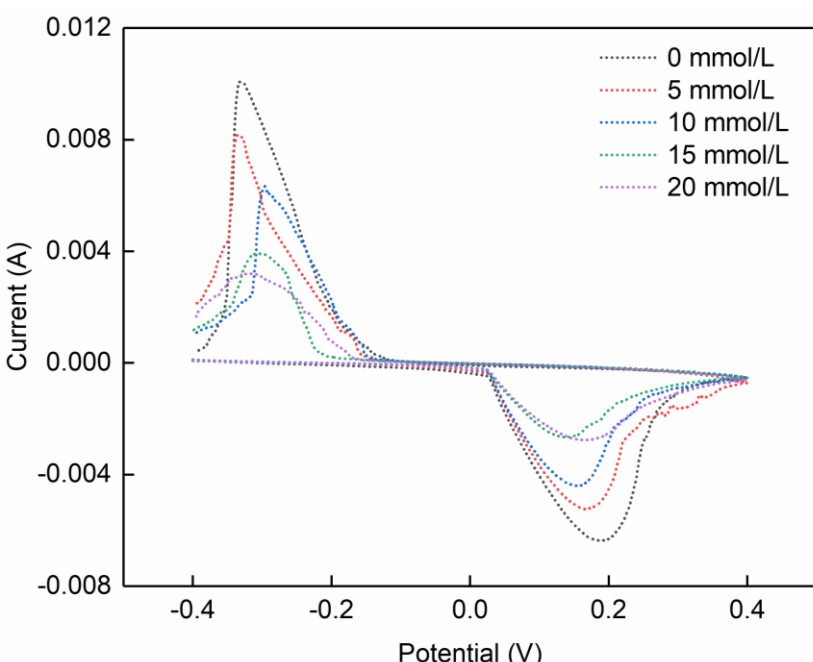

**Figure 8.** The CV curves of three-electrode electrochemical biosensors for glucose detection.

The current difference obtained was calculated using NPPCC and plotted against the glucose concentration to find the correlation between glucose concentration and current response, as shown in Figure 9. Equation (6) was obtained based on the glucose response at different concentrations:

$$NPPCC = 0.0323C + 0.0410 \tag{6}$$

where *C* denotes the glucose concentration.

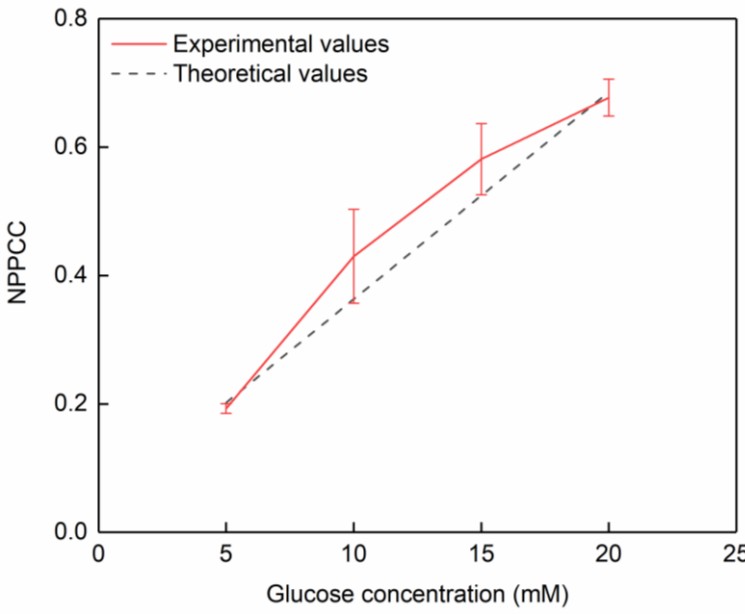

**Figure 9.** The correlation between glucose concentration of current response.

The glucose detection kit consists of a box body manufactured by 3D printing and a detection chip manufactured based on the process method in this study. The detection chip consists of three vertical electrodes, which are a reference electrode, a working electrode, and a counter electrode. The liquid injection port is designed above the box to add the

liquid to be tested. The detection requires only a small amount of the liquid to be tested due to the small size of the designed box, as shown in Figure 10.

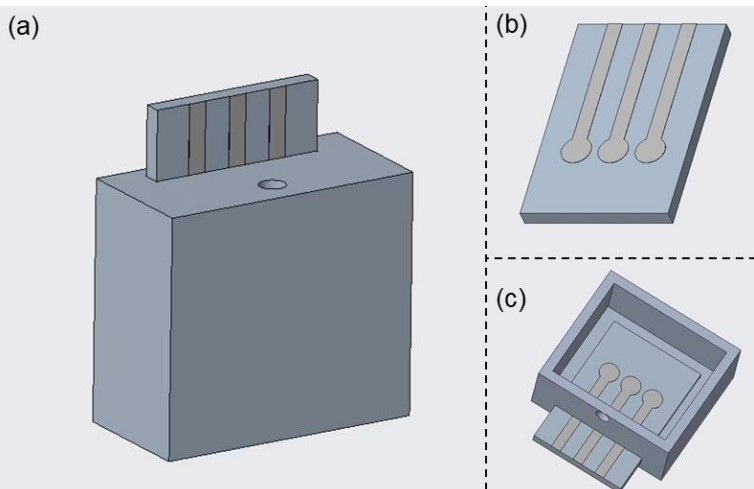

**Figure 10.** The glucose detection kit: (**a**) overall appearance; (**b**) detection chip with three silver electrodes; (**c**) internal structure of the kit.

In the actual inspection process, the chip is inserted into the box through the groove. A small amount of urine is added to the cartridge body through the injection port until the three electrode ends are submerged in urine. Connect the electrodes to the electrochemical workstation to complete the detection. The chip is directly taken out from the groove and can be reused after being sterilized after the test is completed, and the box body is a disposable consumable. The entire detection process can be completed within 5 min, realizing rapid detection.

In this work, the sensor chip is designed on a flat surface. In the future, 3D structures with patterned flow channels can be directly printed at the photocuring processing stage. In addition, this chip is expected to be applied to the detection of various body fluids (such as blood, sweat, tears, etc.).

### *3.4. Repeatability and Reproducibility of the Sensor*

Repeatability and reproducibility were used to judge the performance of the sensors. The repeatability of the sensors was achieved by repeated detection of 10 mM glucose at the NaCl solution. The results showed that the repeatability of the sensors gave relative standard deviation value (RSD) of peak current 8.4%. Besides, the reproducibility of the sensors was checked at room temperature. The data showed 7.6% of RSD value, and that the repeatability and reproducibility of the sensors needed to be enhanced. Thus, the morphology and chemical composition of the sensors were compared, indicating that the chemical composition changes should be further improved to enhance the stability and reproducibility of the sensors.

### 4. Conclusions

In this study, the three-electrode electrochemical biosensor was fabricated by a facile multi-step. Firstly, the detection chips were prepared by DLP, and the Nd: YAG laser activated the copper pattern. The copper pattern was discontinuous and non-conductive, thus the copper pattern provided base points for the electroless plating. Then, the copper and silver coatings were prepared by electroless plating. Finally, the silver electrodes were validated for the effective detection of glucose concentrations in body fluids, and a novel glucose detection kit was designed for convenient detection. The main conclusions are as follows:

> (1) $Cu_2(OH)PO_4$ was added to the resin, and the composite resin could be efficiently fabricated for the detection chips and detection kits by DLP.
> (2) The composite resin was activated by the Nd: YAG laser to form copper base points on the composite resin surface.
> (3) The continuous and conductive copper and silver coatings were prepared by electroless plating. The three-electrode chips were suitable for glucose detection according to the CV and NPPCC.

**Author Contributions:** Conceptualization, R.Z. and L.S.; methodology, F.L.; software, Y.C.; validation, R.Z., Q.W. and Q.Z.; formal analysis, R.Z. and Y.C.; investigation, C.J.; resources, L.S.; data curation, J.X.; writing—original draft preparation, R.Z. and Q.W.; writing—review and editing, Y.C. and C.W.; visualization, J.Z.; supervision, L.S.; project administration, C.W.; funding acquisition, L.S. All authors have read and agreed to the published version of the manuscript.

**Funding:** This research was funded by the Key Research and Development Program of Jiangsu Provincial Department of Science and Technology of China, grant number BE2019002.

**Acknowledgments:** The authors would like to thank the Summer Undergraduate Internship in the Institute of Additive Manufacturing (3D Printing), Nanjing University of Aeronautics and Astronautics, China, for providing laboratory and testing technology.

**Conflicts of Interest:** The authors declare no conflict of interest.

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
