# Peer review of "Biosensors Fabricated by Laser-Induced Metallization on DLP Composite Resin"

_electronics, doi:10.3390/electronics11193254_

Round 1

Reviewer 1 Report

In the manuscript "Biosensors fabricated by laser-induced metallization on DLP composite resin", the authors presented an electrodeless method for producing three-electrode biosensors. Specifically, the authors introduced their fabrication method and used various characterization were performed to validate the material composition. A proof of concept glucose sensing was also performed to assess the sensing ability of their product. Overall, this reviewer would recommend the publication of this manuscript after the below minor revisions.

1, Full description of GOD should be provided.

2, The authors should provide the chemical reaction formula for the electrodeless plating.

3, This reviewer recommends the authors improve the data presented in Figure 8 and Figure 9 with averaged value and standard deviation..

Author Response

In the manuscript "Biosensors fabricated by laser-induced metallization on DLP composite resin", the authors presented an electrodeless method for producing three-electrode biosensors. Specifically, the authors introduced their fabrication method and used various characterization were performed to validate the material composition. A proof of concept glucose sensing was also performed to assess the sensing ability of their product. Overall, this reviewer would recommend the publication of this manuscript after the below minor revisions.

Overall response: We would like to thank you for your careful reading, helpful comments, and constructive suggestions, which has significantly improved the presentation of our manuscript. We have carefully considered all comments from the reviewers and revised our manuscript. Revised portions were marked in Yellow in our revised manuscript.

Point 1: Full description of GOD should be provided.

Response 1: Thank you for your rigorous consideration. We have added the description of GOD in the introduction.

Point 2: The authors should provide the chemical reaction formula for the electrodeless plating.

Response 2: Thank you for pointing out this problem in the manuscript. We felt sorry for our negligence of the chemical reaction formula. We have added them in the revised manuscript.

Point 3: This reviewer recommends the authors improve the data presented in Figure 8 and Figure 9 with averaged value and standard deviation.

Response 3: Thank you for your valuable suggestion. We have revised Figure 9 in the manuscript. However, the data in Figure 8 was obtained from the CV test, thus, averaged value and standard deviation were not applicable.

Reviewer 2 Report

  1. Authors need to give the complete forms of the abbreviations at the first appearance, both in the abstract and introduction.
  2. The authors need to give details about the chemicals used and their procurement in the materials section. 
  3. In Figure 8, why were the CV curves obtained at negative potential values? Explain.
  4. Please verify equation (1) in the manuscript. Why is it not balanced?
  5. Cite the below references for electrochemical sensors in the introduction part.

https://doi.org/10.1088/2053-1591/ab4b92

https://doi.org/10.1016/j.microc.2020.105441

https://doi.org/10.1016/j.eti.2020.101222  

      6. Include a table comparing the analytical performance of glucose biosensors.

  1. Include a separate section on stability. and reproducibility of the developed biosensor.
  2. Insert a figure showing the voltammograms of glucose with/without interferences.
  3. Check and correct the typographical and grammatical errors.

Author Response

Overall response: We gratefully thanks for the precious time the reviewer spent making constructive remarks. In this work, we were committed to the preparation of a proof of concept biosensor that can be used for urine detection. Thus, many experiments were carried out and several characterizations were used to verify the possibility of fabricating biosensors on DLP composite resin. We have revised the manuscript in response to the valuable comments made by the reviewer. Revised portions were marked in Yellow in our revised manuscript. Besides, we noticed that the reviewer made constructive remarks on sensors. We really appreciated the professional advice on sensors, however, we felt sorry that the relevant content could not be provided in this manuscript because we wished to demonstrate a facile approach to fabricate biosensors. According to the references provided by the reviewer, we have carefully studied the relevant theories and hope to gradually improve them in the following work.

Point 1: Authors need to give the complete forms of the abbreviations at the first appearance, both in the abstract and introduction.

Response 1: Thank you for your careful check. We have checked and revised the abbreviations in the abstract and introduction.

Point 2: The authors need to give details about the chemicals used and their procurement in the materials section.

Response 2: Thank you for pointing out this problem in the manuscript. We have provided the details about the chemical used and manufacturing information in the manuscript.

Point 3: In Figure 8, why were the CV curves obtained at negative potential values? Explain.

Response 3: Thank you for your valuable suggestion. Firstly, CV curves can be obtained at negative potential values and positive potential values because the reaction on the reversible electrodes is reversible. Then, CV is a vital method to study the electrode reversibility, and single metal electrodes are usually reversible electrodes. Thus, the silver electrode is a reversible electrode.

Point 4: Please verify equation (1) in the manuscript. Why is it not balanced?

Response 4: Thank you for your rigorous consideration. We felt sorry for the inconvenience brought to the reviewers. We updated the chemical formulas in the revised manuscript.

Point 5: Cite the below references for electrochemical sensors in the introduction part.

https://doi.org/10.1088/2053-1591/ab4b92

https://doi.org/10.1016/j.microc.2020.105441

https://doi.org/10.1016/j.eti.2020.101222

Response 5: We gratefully appreciate for your valuable suggestion in the expertise field of sensors. We have cited them in the introduction, and carefully learnt a lot from the above articles. Besides, we will add relevant contents in the following work and hope these can improve our work.

Point 6: Include a table comparing the analytical performance of glucose biosensors.

Point 7: Include a separate section on stability. and reproducibility of the developed biosensor.

Point 8: Insert a figure showing the voltammograms of glucose with/without interferences.

Response 6 - 8: Thank you for the above suggestions. Firstly, we highly respected the reviewer’s suggestions to test the analytical performance, stability and anti-interference ability of the sensor. We planned to conduct comprehensive work in combination with the above references in the following work to further demonstrate the reliability of this biosensor. Secondly, we focused on achieving preliminary biosensor functions on 3D printed components in this work. Thus, a metallization hybrid processing methods on the DLP resin surfaces was introduced. Silver coatings were successfully prepared on the comnposite resin surfaces. The SEM, EDS, XPS, etc. were carried out to verify the morphological and compositional changes. Finally, CV tests were carried out to prove that the sensor had the ability to discriminate different concentrations of glucose solutions.

Point 9: Check and correct the typographical and grammatical errors.

Response 9: Thank you for your careful check. We have thoroughly checked and corrected the grammatical errors and typos we found in this manuscript.
